# The Interplay of Rites and Customs: The Evolution and Regional Propagation of the Religion of Crown Prince Zhaoming

**Tinglin Sun**

School of Humanities, Guangzhou University, Guangzhou 510006, China; suntinglin@126.com

**Abstract:** Previous studies on the religion (*xinyang* 信仰) of Crown Prince Zhaoming 昭明太子, focused on the welcome ceremony of the Nuo deities 儺神 (the deities driving away the plague) and the historical figure of Crown Prince Zhaoming but on the other hand, overlooked the evolution from the folk or heterodox deity Jiulang Shen 九郎神 (Jiulang God) to the state-recognized or orthodox deity Crown Prince Zhaoming and the role of Song-era national policy of conferring titles and inscriptions upon deities in this process. This paper aims to illuminate the following five points: Firstly, the original deity upon which the religion of Crown Prince Zhaoming in Chizhou 池州 is based is Jiulang God, one of many deities in the Nuo religion imbued with rich elements of Wuism (*wuxi* 巫覡). The religion of Jiulang ascended to its peak during the Tang dynasty (618–907). Secondly, driven by the discourses of scholars, government officials and national rites during the Song dynasty (960–1276), Jiulang was transformed into Crown Prince Zhaoming through the conferment of titles and inscriptions, becoming an orthodox deity. Thirdly, Crown Prince Zhaoming and Jiulang God coexisted for a prolonged period, and this suggests that the rites–customs dichotomy was universally found in folk religions in traditional China. Fourthly, the proliferation of the religion of Crown Prince Zhaoming in western Anhui, Jiangxi, Jiangsu, Zhejiang and other regions of China reveals a mix of factors that led to the widespread and lasting prevalence of the religion. These factors include the deity's role as a guardian of maritime voyages and merchants; a stable, enduring organizational structure for sacrificial rituals; and the dichotomous coexistence of rites and customs. This article reveals that from the Song Dynasty, the national system of rites permeated and impacted folk religions through official and academic discourses, propelling the latter's continuing transformation into an orthodox form. Nevertheless, there remained spaces between the national system of rites and folk religions where "rites" and "customs" interacted, integrated and coexisted well. The interplay, fusion and peaceful coexistence between "rites" and "customs" is the normative state of folk religions in traditional society in China, as well as one of the key reasons why many folk religions continue to flourish and play a societal role.

**Keywords:** rites–customs dichotomy; Crown Prince Zhaoming; folk deities; national ritual deities

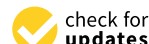



## 1. Introduction

Kristofer Schipper highlighted the differences between Taoist priests in Taoism and folk religions (Kristofer Schipper 1985, pp. 21–57). In his studies of folk beliefs in China, J. Watson in his article titled "*Standardizing the gods: the promotion of T'ien Hou ('Empress of. Heaven') along the South China coast, 960–1960*" analyzes how the state and elites standardized and transformed the T'ien Hou (also known as Mazu) cult into a state religion (Watson 1985, pp. 292–324). Valerie Hansen considers the dissemination of folk beliefs to be a manifestation of the commercial revolution in the religious world in the Song dynasty, and that the creation and spread of local deities were closely related to the development of merchants and commerce (Hansen 1990, p. 3). Edward Davis demonstrates the renewed vitality of Taoist, Buddhist and local religious traditions in the Song dynasty (960–1276),

when folk beliefs underwent an uninterrupted transformation driven by Confucianism, Buddhism and Taoism (Edward L. Davis 2001). Vincent Goossaert details the implications of the notion of 'religion' in modern and contemporary China and looks at how Chinese religious traditions have adapted to it (Vincent Goossaert 2005, pp. 13–20). David W. Faure and Zhiwei Liu examine the diversity of local cultures as reflected in rituals and show it is connected to the perceived unity of Chinese culture, and they hold that documenting the history of the adoption of legitimizing symbols can help to better the complex history of the making of the perceived unity of Chinese culture (Faure and Liu 2008, pp. 1–21). In addition, "great tradition vs. little tradition" and "nation vs. region", amongst others, are approaches that are commonly employed in the study of folk religions. A lot of attention is placed on oppositional dichotomies. Comparatively, studies on the middle ground that exists in between or the fluidity with which the extremes transition to the other—in other words, the space in which folk religions survive and thrive—are fewer. Take studies on the religions of deities during the Song era. An excessive emphasis on the delineation between orthodoxy (*zhengsi* 正祀) and heterodoxy (*yinsi* 淫祀) can easily lead one to neglect the vast "middle ground" that exists between the two (Pi 2008, pp. 208–24). Through his study on the evolution of the main deities worshiped at Jinci Temple in Taiyuan, Zhao Shiyu 赵世瑜 concluded that the division and fusion of rites and customs in folk religion had an impact on the metamorphosis of deities (S. Zhao 2015, pp. 10–12). Zhang Shishan 张士闪 suggested that the interplay between rites and customs in traditional Chinese society interlinked the nation and the populace while cultural recognition led to the elimination of existing and potential societal threats, leading to the establishment of a foundation upon which the drafting of national policies and functioning of society rested (Shishan Zhang 2016, pp. 14–24). The interplay between rites and customs has become a key approach through which one conducts research and explores folk religions in traditional society in China. The above studies provide multiple research perspectives for the study of Chinese folk beliefs, and it has gradually become a consensus that, in the studies of Chinese folk religions, employing a multi-disciplinary approach for more in-depth case studies can facilitate a deeper understanding of the field, which, in turn, can lead to theoretical breakthroughs. (L. Liu 2009, pp. 90–104; Pi 2009, p. 75). The objective of this paper is to straighten out the origins of the religion of Crown Prince Zhaoming.

The beliefs in the religions of Crown Prince deities prevailed in the Jianghuai region; among them, the religion of Crown Prince Zhaoming was prevalent in many places in Jianghuai and Jiangnan with Chizhou as the center, and it was one of the important folk religions after the Tang and Song dynasties. Yu Zhengxie, a scholar in the Qing Dynasty, distinguished the belief of Tongzhen Third Prince 通真三太子 and Crown Prince Zhaoming 昭明太子 in *Guisi Cungao* 癸巳存稿 (Yu 2003, vol. 13, p. 407). Currently, the latter is viewed as a Nuo deity or a local deity in a majority of relevant studies (Z. Wang 1997, pp. 51–62). Due to the eminent position that the historical figure Crown Prince Zhaoming, or Xiao Tong, holds in literary history, some scholars examined Crown Prince Zhaoming as a deified personality in their studies on the religion and its influence on the region and community (Fang 2011) while defining related folk customs as momentous intangible cultural heritage (Tan 2019, pp. 55–59). However, comprehensive studies are absent in regard to the origin and evolution of this religion, especially regarding Jiulang God which was a heterodox Nuo deity—i.e., how it was transformed into the famous figure Crown Prince Zhaoming under the national rites system as well as how the state-recognized God Crown Prince Zhaoming and the folk god Jiulang God interacted, fused and coexisted for a long time.

The compilation of *Zhaoming Taizi Shishi* 昭明太子事实 (*Facts About Crown Prince Zhaoming*) by Zhao Yanbo 赵彦博, the magistrate of Chizhou, during the Qiandao reign (1165–1173) of the Southern Song dynasty is the earliest compilation of information related to the religion. The book was compiled during the Qiandao reign and reprinted in the Chunyou reign. Records of the compilation can be found in *Zhizhai Shilu Jieti* 直斋书录解题 (*Catalog of Zhizhai Study*), *Wen Xian Tong Kao* 文献通考 (*Verifying Documents*), and *Songshi:*

*Yiwen Zhi* 宋史· 艺文志 (*History of Song: Catalog of Literature*). The listing of *Facts About Crown Prince Zhaoming* alongside *Cishan Jiashi Bian* 祠山家世编 (*Cishan's Genealogy*) and *Haishen Lingying Lu* 海神灵应录 (*God of Sea's Prophecies*) in the *Wen Xian Tong Kao* 文献通考 (Ma 1986, vol. 199, p. 1668), in particular, categorized it as a book on folk religion. The only surviving copy of *Facts About Crown Prince Zhaoming* is currently in the collection of Qing-era manuscripts housed in the National Library of China. By focusing on the book *Facts About Crown Prince Zhaoming*, this author intends to identify how Jiulang God, a folk or heterodox deity, was transformed into Crown Prince Zhaoming, a state-recognized or orthodox god, taking into account the role of the Song-era conferments of titles and inscriptions upon deities and the narrative espoused by scholars and local officials under the national rites system. The coexistence and interplay of the dual representations of the deity have facilitated the popularity of the religion of Crown Prince Zhaoming and its proliferation to the vast regions of Jianghuai and Jiangnan. This study aims to enhance people's understanding of the general principles that determine the evolution and underlying mechanisms of folk religion in traditional society in China.

## 2. The Origin: Jiulang God, a Heterodox Nuo Deity

Since the Song dynasty, the worship of Crown Prince Zhaoming has proliferated in the vast Jiangnan region. According to many records, the deity was originally known as Jiulang (九郎, the ninth son). In the fourth year of the Xuanhe reign (1122), Huang Yanping 黄彦平, a man from Fengcheng, Jiangxi, paid respects at the deity's temple and later wrote in *Wenxiao Miao Ji* 文孝庙记 (*On Wenxiao Temple*) that the deity's followers originally addressed him as Jiulang (Yanping Huang 1986, vol. 4, p. 781). In his travelogue, Lu You 陆游 of the Southern Song dynasty noted as he passed Chizhou that Jiulang God resided in western Chizhou, who answered prayers, with some saying that Jiulang was Zhaoming (Lu 1985, vol. 3, p. 25). According to Zhou Bida 周必大 (1126−1204), there was a grand temple located in western Chizhou, for a deity whose followers were called Guoxi Jiulang (*Guoxi* 郭西 literally means the west of a city) (B. Zhou 1986, vol. 169, p. 835). Similarly, *Yudi Jisheng* 與地纪胜 (*Exhaustive Description of the Empire*) documented a temple for Crown Prince Zhaoming of the Liang dynasty (502–557) found outside Chizhou's Qiupu Gates. The deity in the temple was referred to by the people as Guoxi Jiulang (X. Wang 2005, vol. 22, p. 1054). These records demonstrate that officials, scholars and writers referred to the deity as "Crown Prince Zhaoming" while the common populace continued their age-old tradition and called their deity "Jiulang".

Crown Prince Zhaoming is a historical figure. The eldest son of Emperor Wu of Liang (梁武帝) who died at a young age. According to *Liang Shu* 梁书 (*Book of Liang*), "men and women running through the gates of the palace and wailing in the streets as well as commoners across the land and near the borders mourning" when he passed (Yao 1973, vol. 8, p. 171). However, this is simply an embellishment in the historical records. In reality, neither the imperial government nor the people erected a temple for the late crown prince then. Later, he was posthumously granted the title of Emperor Zhaoming with the temple name (庙号, a posthumous title of an emperor) Gaozong (高宗) by his son Xiao Cha (萧詧) after the latter acceded to the throne in Jiangling 江陵 (Linghu 1971, vol. 48, p. 859). However, the ruling Xiao clan had done so to observe official rites. The conferment had nothing to do with popular folk religion.

Jiulang had been a popular figure of worship in folk religion in the Jianghuai region. During the early Song dynasty, Qian Yi 钱易 documented in *Nanbu Xinshu* 南部新书 (*New Southern Anecdotes*) the prevalence of Jiulang Temples (九郎庙) and General Mao Temples (茅将军庙) in Jianghuai. Jiulang, also known as the ninth son of Fu Jian 苻坚 (Emperor of Former Qin), could sense soldiers of the underworld, answered many prayers and performed many miracles (Qian 2003, p. 93). Detailed records on Jiulang are scarce. Most records classified General Mao as a heterodox deity (J. Li 2002, vol. 5, p. 239). During the deity's summoning, "the shamans run amok, and fortune and misfortune become entangled" (Z. Li 2017, p. 54). This shows that the religion of Jiulang had flourished since

the period of the Five Dynasties period (907–960) and the early Song. In addition, the religion was comparable to the religion of General Mao, both being recognized as heterodox religions with rich Wuism elements.

The folk rites of worship of Jiulang during the Song dynasty were grand. "The people here conduct worship with genuine respect and do their best in the activity, regardless of being rich or poor, wise or foolish. The mid-autumn ritual of worship is a yearly affair. The rites are complicated, elaborate, and costly. Everyone performs them eagerly and tries to outmatch one another (Y. Zhao n.d., Juanxia 卷下)." As a religion with a long history and a stable pool of believers, its rituals of worship remained quite consistent. As a result, a few things can be discerned about the early forms of the religion through the study of its sacrificial rituals in the Ming and Qing eras. Wu Yingyan 吴应箕, who was born in Guichi 贵池 and lived during the late Ming and early Qing eras, detailed the welcome ceremony for the deity in *Chizhou Ying Zhaoming Hui Ji* 池州迎昭明会记 (*Account of Chizhou's Welcome Ceremony for Zhaoming*):

The elders perform a sacrifice at the temple on the first day of the eighth month. The welcome ceremony for the deity on the twelfth day of the eighth month is especially grand. Prefectural and county officials escort the deity from Guoxi Temple to Zhusheng Temple (Jingde Temple) in Chizhou . . . Crown Prince Zhaoming's birth fell on the fifteenth day of the eighth month . . . The welcome ceremony for the deity is conducted that day. Members of the reception dress up as Guan Gong 关公 (Lord Guan), Cheng Huang 城隍 (City God), Qi Sheng Erlang 七圣二郎 (Er-lang and the Seven Sages), and Yuanshuai 玄坛元帅 (Lord Zhao the Marshal). Everyone rides on a horse and wears a mask. They are donned in elaborate costumes and accompanied by escorts. The Seven Sages are the exception. They employ mechanisms that have blades piercing their neck and abdomen as they walk behind flags and a marching band. The statue of the deity Crown Prince Zhaoming is carried on a sedan as the procession parades the streets and travels to the prefectural and county courts. The parade ends in the evening . . . In western Chizhou lives a deity called Jiulang . . . Perhaps the deity is numbered the ninth (Jiu) because that is the sum of two (Er) in Erlang and seven (Qi) in Seven Sages . . . Perhaps the deity shares a brotherhood with the Seven Sages and Erlang subdued him with two deities (Lang 2002, vol. 9, p. 734).

The association between "Erlang and the Seven Sages" and "Jiulang" is noteworthy. The Seven Sages, who "employ mechanisms that have blades piercing their neck and abdomen", are represented in a strikingly different manner when compared to the other deities such as Lord Guan and the City God. Hypotheses postulating that "the deity is numbered the ninth (Jiu) because that is the sum of two (Er) in Erlang and seven (Qi) in Seven Sages" or that "the deity shares a brotherhood with the Seven Sages" reflect how "Jiulang" remained the popular representation of the deity in folk religion during the Ming and Qing eras—one that is markedly contrary to scholars' depiction of the deity as Crown Prince Zhaoming, or Xiao Tong, a historical figure renowned for his filial piety and literary legacy, in academic discourse. Supporting the above are records by Liu Tingluan 刘廷銮, who was born in Guichi and lived during the late Ming and early Qing eras, about Shidai County 石埭县: The people of Shidai worship Chang Shen 猖神, who answers their prayers readily. Every fifteenth day of the eighth month, they hold a welcome ceremony for Chang Shen and vie to invite the deity into and enshrine him in their homes, then send him off during the ceremony that is held the following year. There are five ancient Chang Shen statues. Hosts must acquire the approval of local officials before they are permitted to invite the deity into their homes. On the day of the reception, a sacrifice is to be made at the Chang Shen temple. A chicken must be slaughtered next to the censer. The people believed that Chang Shen is Crown Prince Zhaoming (T. Zhou 1675, vol. 8).

Liu documented the event on the fifteenth day of the eighth month in *Shidai Zhongqiu Ci* 石埭中秋词 (*Shidai Mid-autumn Poem*). The "*Liang Di* 梁帝 (*Emperor of Liang*)" in his poetry he refers to Crown Prince Zhaoming while descriptions of "temple drums, silver lights, performances", "red wigs and black masks", "slaughtered chicken and their rivers of blood", and "men wielding swords and riding donkeys" paint the picture of a grand af-

fair. Religious followers "burn incense late at night", "provide offerings of pleasing food", and "fight to get the first red temple seal stamp" (T. Zhou 1675, vol. 8). Their behavior reveals the immense sway that the deity had over the common people. In conclusion, what scholars documented as "Crown Prince Zhaoming" in written records is the Jiulang that the common folk worshiped in practice. The former led to the creation of a Jiulang that is in the image of Crown Prince Zhaoming.

Song-era Chizhou was under the administrative jurisdiction of Jiangnan East Circuit and was adjacent to Jiangnan West Circuit. The people in the region were heavily influenced by Wuism. During the period, local people "venerated spirits and followed ancient Wuism customs". When welcoming deities, they "created sculptures and paintings of spirits, displayed flags, and played drums and horns", among other things (Xu 2014, p. 992). Such behavior resembles the celebratory events welcoming "Zhaoming" that were held during the Song and Ming eras, which were mentioned earlier in this paper. In view of the above, this author believes that, prior to the Song dynasty, the religion of Crown Prince Zhaoming was already thriving in the Jianghuai region. The deity was initially called Jiulang and of a Nuo religion that was heavily influenced by Wuism.

The Nuo religion has a long history, and its primary function is to expel ghosts and plagues (Rao 1993, vol. 6, p. 32). The religion of the Nuo deity Jiulang is of similar antiquity. There are three key years or periods that were mentioned in records written during the Song dynasty. The first—documented by then Chizhou's administrator for revenue and taxation Zhong Shimei 钟世美 in *Miao Ji* 庙记 (*On Temples*)—is the first year of the Yuanyou reign (1086). It was believed that the people of Chizhou began worshiping Crown Prince Zhaoming after his death. When the city's administrative center was relocated in the first year of the Yongtai reign of the Tang dynasty (765), the people of Chizhou relocated the temple from Xiushan to the west of Chizhou. The religion of Crown Prince Zhaoming began to spread across Jiangdong (Y. Zhao n.d., Juanxia 卷下). The second—put forth and documented by Zhang Bangji 张邦基 in *Mozhuang Manlu* 墨庄漫录 (*Mo Villa Files*)—is the Kaicheng reign (836−840) during the Tang dynasty and during which the Zhou clan began taking charge of the sacrificial rituals for Crown Prince Zhaoming. The clan continued to preside over the affair until the Song era (Bangji Zhang 2008, vol. 4, p. 46). The third is the third year of the Duanping reign (1236) during the Southern Song dynasty. Then assistant to the prefectural chief and Qingyang native Ye Zhi 叶寘 (who went by the pseudonym Tanzhai) claimed in *Daoyu Ganying Ji* 祷雨感应记 (*Notes on Praying for Rain*) that Crown Prince Zhaoming was worshiped in Chizhou and the practice began in the Tianbao reign (742−756) during the Tang dynasty (Y. Zhao n.d., Juanxia 卷下).

These three claims appeared during the Song era, which was a key period that marked Jiulang's metamorphosis into Crown Prince Zhaoming. That explains why Jiulang was referred to as Crown Prince Zhaoming in the aforesaid three writings. Two things can be discerned from them. Firstly, in the first year of the Yongtai reign (765), the nexus for the religion of Jiulang shifted from the ancestral temple in Xiushan, Chizhou to the temple in Guoxi (west of) Chizhou. Followers began to flock to the temple in Guoxi. The fame of Guoxi Jiulang and Guoxi Temple spread far and wide, driving the proliferation of the religion across Jiangdong. Secondly, between the Kaicheng reign (836−840)—when the Zhou clan (周氏) began its intergenerational charge over ancestral worship at the temple—and the early 12th century, when "the descendants of the Zhou clan split into eight factions, for the best" (Bangji Zhang 2008, vol. 4, p. 46), a consistent religious staff, including the acolyte, and a comprehensive set of ancestral rites, was established. The religion's organizational structure reached a certain level of maturity and stability.

In conclusion, this author is of the opinion that the religion of Crown Prince Zhaoming was derived from the religion of Jiulang God. The religion of Jiulang God was a Nuo religion with rich elements of Wuism, as well as a heterodox religion alongside the religion of General Mao. During the Tang dynasty, the central place for the worship of Jiulang God shifted from the ancestral temple in Xiushan to the temple in western Chizhou. The period marked the Zhou clan taking charge of sacrificial rituals for several generations and was

a key period during which the religion of Jiulang God ascended to its peak. Up until the 10th century, the religion of Jiulang God flourished in the Jianghuai region. Claims of the deity being "Fu Jian's ninth son", able to communicate with "soldiers of the underworld", and others were widespread among the populace. During this period, the deity had not yet been associated with Crown Prince Zhaoming.

### 3. From "Customs" into "Rites": The State-Recognized Deity Crown Prince Zhaoming Replacing the Folk Deity Jiulang

Since the beginning of the Northern Song dynasty (1127–1279), the portrayal of Jiulang has been steadily refashioned into the image of Crown Prince Zhaoming in official and academic discourse. Zhong's *On Temples*, dated the first year of the Yuanyou reign (1086), is the earliest surviving text and thus worth greater, detailed study. In *On Temples*, it was written:

> According to both local history and folklore, Crown Prince Zhaoming named the place Guichi after he had a taste of the delicious fish in Qiupu County 秋浦县, Chizhou. After his death, the crown prince's spirit possessed a commoner in Xiushan, Chizhou and spoke through the commoner, "I favored Chizhou while I was alive. God has bestowed upon me the city. Worship me and I shall protect and bless you." Then, local people built a temple in Xiushan, Chizhou and performed rituals of worship in autumn and winter. They would pray whenever the need arose and their prayers were always answered. They were the envy of the common folk and the wealthy. In the first year of the Yongtai reign of the Tang dynasty, the prefecture was established next to the Yangtze River. The people of Guichi built the temple in the west of the city walls. Temples were found all across Jiangdong. (Y. Zhao n.d., Juanxia 卷下)

The text included the claim that after Crown Prince Zhaoming died, his spirit appeared in Xiushan. As a result, the people built a temple in his name and began worshiping him. That is how Jiulang acquired the association with Crown Prince Zhaoming. Similar claims and interpretations can be traced back to the Jiayou reign (1056−1063). Xia Jiang, a man from Chizhou, asserted in his writing that Guoxi Jiulang was formerly from Chizhou. During the Jaiyou reign, Guichi's magistrate Wang Ji 王霁 concluded that Jiulang was Crown Prince Zhaoming after consulting illustrated texts (Y. Zhao n.d., Juanxia 卷下). Therein lies the question worth examining—why were local officials such as Wang and Zhong so certain that Jiulang was Crown Prince Zhaoming?

Records associating the naming of Chizhou with Crown Prince Zhaoming have existed as early as the Southern Dynasties period (420–589). In *Yudi Zhi* 輿地志 (*Geographical Anthology*), Gu Yewang claimed that Crown Prince Zhaoming named the waters Guichi, after the good fish that they breed (F. Li 1960, vol. 170, p. 827). While Gu drew an association between the naming of "Guichi" and Crown Prince Zhaoming, he made no mention of the crown prince having visited the place before. Such a practice of associating places with historical figures was commonly used in the shaping of regional culture during the Southern Dynasties period. Such accounts are documented in local illustrated or geographical texts and cited in other texts such as *Yuanhe Junxian Zhi* 元和郡县志 (*Geographical Index of Counties in the Yuanhe Reign*) and *Taiping Huanyu Ji* 太平寰宇记 (*Universal Anthology in the Taiping Reign*). They are cultural resources upon which regional histories and traditions were built and that were continually enriched with successive interpretations. Similar to the *Geographical Anthology*, which stated that Crown Prince Zhaoming named Guichi "after the good fish that they breed" and contained no mention of the crown prince ever visiting the place, the *Universal Anthology in the Taiping Reign*, too, cited "the good fish that the waters breed". Such statements were further interpreted, leading to the conclusion in local texts of the Ming and Qing eras that Crown Prince Zhaoming had personally visited Chizhou. The cultural resources that were formed from such interpretations became the historical evidence upon which local officials of Chizhou drew during the Song dynasty to arrive at the conclusion that Jiulang is Crown Prince Zhaoming.

Although Crown Prince Zhaoming never actually visited Chizhou, the official literati invented the story of his spirit appearing in Xiushan in Chizhou after Crown Prince Zhaoming's death. The construction of this narrative led to the undermining and erasure of Jiulang's original image and is the reason why poetry and texts written by later generations contained claims of "Crown Prince Zhaoming appearing as a spirit in Chizhou after his death". Yet, the fabrication is not flawless. Crown Prince Zhaoming, or Xiao Tong, had never visited Chizhou. He was the eldest son of Emperor Wu of Liang—there is no reason why he was called "Jiulang" (the ninth son) at all. This is why Zhong had to plug the logical gap with an explanation on the relation between "Crown Prince Zhaoming" and "Jiulang" in *On Temples*—by drawing a connection between the filial piety and love for his people that Crown Prince Zhaoming exhibited through the deeds he performed while he was alive and the "virtue" that Jiulang displayed by protecting the people and "answering their prayers" (Y. Zhao n.d., Juanxia 卷下).

At this juncture, the reconstruction of Jiulang's image as Crown Prince Zhaoming was almost complete. Several titles were conferred upon the deity successively as local cultures and traditions were assimilated into national rites during the Song dynasty. According to *Chici Wenxiao Miao'e* 敕賜文孝廟額 (*Imperial conferment of Inscription on Wenxiao Temple*), dated the third year of the Yuanyou reign (1088):

Ministerial records on Wenxiao Temple in Chizhou from the Ministry of Rites read: Request of Chizhou redelivered from Jiangnan East Circuit: "The (Chizhou) prefecture received little rainfall last summer. Prayers made at the temple for Crown Prince Zhaoming of Liang west of the city walls have been answered readily. The temple's location is noted on the map in the illustrated local chronicles. We hereby request the conferment of a title upon the deity. The Ministry has sought assistance from the Court of Sacrificial Worship. After a detailed examination, the Court proposes to bestow an imperial inscription and title of Wenxiao" (Y. Zhao n.d., Juanxia 卷下).

Chizhou's local authorities requested the central government for an imperial title and inscription. Named explicitly in their request is "Crown Prince Zhaoming"; missing in the request is the folk name "Jiulang". The imperial inscription of "*Wenxiao*" 文孝 (Learned, Filial) is not a reference to the deity's ability to answer prayers but to the virtues of the historical figure Crown Prince Zhaoming, or Xiao Tong. A narrative steered by local officials was written in text and subsequently affirmed by the central government through an imperial inscription. The conferment of the title and inscription under the national rites system strengthened the representation of the deity as Crown Prince Zhaoming continually.

Successive conferments of titles and inscriptions from the central government followed. Between the fourth year of the Chongning reign (1105) and the ninth year of the Qiandao reign (1173), five titles were granted, culminating in a king's eight-character title—"英濟忠顯廣利靈佑" (Wise, Charitable, Loyal, Manifest, Far-Reaching, Meritorious, Responsive, Guardian) (Y. Zhao n.d., Juanxia 卷下). According to Song-era rules and regulations on the conferments of titles upon deities, a king's title with eight characters is the highest title that a deity could be granted. On *Gaifeng Wen Xiao Ying* Ji Zhong *Xian Ling You Wang* Gaochi 改封文孝英濟忠顯靈佑王告敕 (Decree on Changing the Title to Learned, Filial, Wise, Charitable, Loyal, Manifest, Responsive, Guardian King) dated the first year of the Jiatai reign, it was noted that "the title no longer befits the noble king, thus a promotion is granted based on public sentiment", henceforth revising the deity's title to "Learned, Filial, Wise, Charitable, Loyal, Manifest, Responsive, Guardian King" (1201) (Y. Zhao n.d., Juanxia 卷下). During the Southern Song dynasty, a scholar called Jing Hao 景暭 "engraved the aforesaid titles on stones and had them placed in the temple" (Y. Zhao n.d., Juanxia 卷下). The imperial decrees on the conferments of titles were thus made known to the masses, and through this process, the state-recognized deity Crown Prince Zhaoming was promoted among the common populace.

Records of Crown Prince Zhaoming's entombment in Xiushan, Chizhou have appeared since the Southern Song dynasty. Take *Ru Shu Ji* 入蜀记 (*Journey to Shu*), in which Lu You wrote: "The tomb of Crown Prince Zhaoming of Liang rests in Xiushan and is sur-



rounded by towering trees." The *Yudi Jisheng* 輿地紀勝 stated that Crown Prince Zhaoming's "tomb is found in Xiushan, Guichi." Similarly, it was written in *Huanyu Tongzhi* 寰宇通志 (*Global Annals*) that "the tomb of Crown Prince Zhaoming is found on Xiushan, west of the city. The crown prince's family name was Xiao, his first name Tong, and his courtesy name Deshi. He was the son of Emperor Wu of Liang (X. Chen 1985, vol. 12, p. 500)." In reality, Crown Prince Zhaoming's tomb is located forty to fifty miles northeast of present-day Nanjing (Xu and Zhang 2015, p. 49). Xiushan is the original location of the ancestral temple for Jiulang and the birthplace of the religion. If there is to be a tomb at Xiushan, it should belong to Jiulang. Since the Jiayou reign of the Northern Song dynasty, scholars had replaced "Jiulang" with "Crown Prince Zhaoming" in their discourse. During the Southern Song era, Jiulang's tomb in Xiushan, Chizhou was referred to as Crown Prince Zhaoming's tomb when it did not, in reality, belong to the historical figure Crown Prince Zhaoming.

From the Song dynasty, increasingly more cultural sights associated with Chizhou and Crown Prince Zhaoming were created. Calligraphy by Crown Prince Zhaoming's hand of "Yinshan Temple" 隐山之寺 can allegedly be found in Jiande County 建德县. In the second year of the Xuanhe reign (1120), Guichi's vice magistrate Zhang Bi 张畀 made a copy of the original calligraphy, engraved it on a stone and enshrined it in the Guoxi Temple in Chizhou. During the Chunxi reign, Chizhou's magistrate Yuan Shuoyou 袁说友 "retrieved the original calligraphy and enshrined it next to the temple" (Wu 1986, vol. 17, p. 500). During the Qiandao reign of the Southern Song dynasty, Chizhou's magistrate Zhao Yanbo compiled *Facts About Crown Prince Zhaoming*. The book was expanded upon and reprinted during the Chunyou reign. This explains why Song's local officials in Chizhou stated, "Little known during the Sui and Tang eras as well as the Five Dynasties period, he rose to prominence in the present dynasty ... The religion reached its height during the Yuanyou reign. A refined hall was built next to the right-wing building, and murals of ceremonial guards were painted on the walls of the corridors. The place has everything, from the tablets of deities, carriages and horses, to a library" (Y. Zhao n.d., Juanxia 卷下). The creation of numerous sights that were related to Chizhou and Crown Prince Zhaoming was steadily completed during the Song dynasty. The alleged ruins in Chizhou that are seemingly associated with Crown Prince Zhaoming were likely the results of local cultural invention since the Song dynasty (Guo 2000, p. 85). Historical artifacts of ancient sages were relocated to other locations or reassembled to form new sights. The transformation of the historical memory of ancient sages into physical sights is the key means through which local history was reconstructed (Lan 2021, p. 80).

In conclusion, during the Jiayou reign (1056–1063), Guichi's magistrate Wang Ji affirmed Crown Prince Zhaoming to be Jiulang. Alongside the involvement of Zhong Shimei and other local officials in the reshaping of the folk deity, the heterodox Nuo deity "Jiulang" was refashioned into the state-recognized deity "Crown Prince Zhaoming". After the Yuanyou reign, building upon the existing foundation and answering the need to indoctrinate the populace, local officials acted in accordance with the national ritual policy of conferring titles upon deities and requested the same for "Crown Prince Zhaoming". The metamorphosis from the folk or heterodox deity Jiulang God to the orthodox deity Crown Prince Zhaoming is closely intertwined with the extensive conferments of titles, integration of folk religions and traditions, and proactive instruction on rites during the Song dynasty. During the period, "most imperial inscriptions and titles were conferred during the Xining, Yuanyou, Chongning and Xuanhe reigns" (Tuqto'a 脱脱 1977, vol. 105, p. 2562). Zhong's *On Temples* was written in the first year of the Yuanyou reign. The first imperial inscription of "Wenxiao" was conferred upon Jiulang in the third year of the same period (1088). Subsequently, three titles were conferred in the fourth year of the Chongning reign (1105), the first year of the Daguan reign (1107), and the first year of the Zhenghe reign (1111) (Xu 2014, 礼 20, pp. 998–99). This period marked the height of the Song dynasty's legitimization of folk deities under the national rites system. It was in this context that Jiulang was transformed through the discourse of local officials and scholars into Crown Prince Zhaoming,

becoming one of the state-recognized deities with titles and inscriptions granted by the state. Endorsed by the state and perpetuated in academic discourse, the legitimate deity Crown Prince Zhaoming continued to strengthen, stabilize and drive the emergence of numerous cultural sights associated with Chizhou and Crown Prince Zhaoming.

## 4. The Coexistence of "Rites" and "Customs": The Interplay and Fusion of the Folk Deity Jiulang and the State-Recognized Deity Crown Prince Zhaoming

The Jiayou reign of the Northern Song dynasty marked Chizhou officials' establishment of an associative relation between Jiulang and Crown Prince Zhaoming while the Yuanyou reign marked the central government's multiple conferments of titles upon the deity. During this period, the national ideology entered the religion of Jiulang through the national rites system and conferments of titles and gradually transformed the Jiulang of Wuism religious roots into the "Crown Prince Zhaoming" in academic discourse. A "pantheon" 万神殿 was created through the Song's national rites system. "Cimiaomen 祠庙门 (Temples)" in *Song Huiyao* 宋会要 (Song Compendium) documented: Wenxiao Temple is located in Guichi, Chizhou. The deity was granted the title of Wenxiao in the fourth year of the Yuanyou reign of Emperor Zhezhong. On the tenth month in the fourth year of the Chongning reign of Emperor Huizong, the deity was granted the title of *Xian Ling Hou* 显灵侯 (Lord Who Answers Prayers). On the sixth month in the first year of the Daguan reign, he was endowed with the title of *Zhao De Gong* 昭德公 (Virtuous Lord), and on the second month in the first year of the Zhenghe reign, the title of *Ying Ji King* 英济王 (Wise, Charitable King). On the third month in the thirtieth year of the Shaoxing reign, Emperor Xiaozong added two characters, *Zhong Xian* 忠显 (Loyal, Manifest), to the deity's title. On the sixth month in the third year of the Qiandao reign, Emperor Xiaozong added the title *Ying Ji Zhong Xian Guang Li Wang* 英济忠显广利王 (Wise, Charitable, Loyal, Manifest, Far-Reaching, Meritorious King) ([Xu 2014](#), 礼 20, pp. 998–99).

With the metamorphosis of "Jiulang God" into "Crown Prince Zhaoming", the deity joined the ranks of past preeminent kings—the likes of Emperor Shun, Yu the Great, Emperor Wu of Wei, Taibo of Wu and Shu Yu of Tang. His temple was classified under "temples for past emperors, kings, and renowned officials" while the deity himself was deemed a former emperor and an ancestral spirit. Records further stated: "Zhaoming Temple is the temple for Crown Prince Zhaoming of Liang. His name is Xiao Tong. He has been conferred a king's title and the posthumous title of *Ying Ji Zhao Lie Guang Li Zhong Xian* 英济昭烈广利忠显 (Wise, Charitable, Virtuous, Accomplished, Far-Reaching, Meritorious, Loyal, Manifest). During the Zhiping reign, the people invited the deity to Chizhou and built a temple there. The temple was named Wenxiao Temple of Xuancheng ([Xu 2014](#), 礼 21, p. 1087)." Be it in Chizhou or other prefectures and counties to which the religion had spread, the deity was identified as a former emperor in official narratives.

Judging from multiple records written during the Song dynasty, both representations of the deity—Crown Prince Zhaoming and Jiulang—continued to coexist during the same period. Including accounts by Lu You and Zhou Bida, that were mentioned earlier in this paper, and by Huang Tingjian 黄庭坚, who passed Chizhou and noted how "the people worshiped Zhaoming, referred to him as Guoxi Jiulang, and attributed the latest drowning of twelve people after a ship capsized to a display of the deity's power (Tingjian [Huang 1986](#), vol. 5, p. 382)." The "Crown Prince Zhaoming", in the official and academic discourses under the national system of rites, and the Nuo deity "Jiulang", in the everyday life of the people, coexisted in a non-contradictory duality. Huang Yanping from Fencheng, Jiangxi revealed the interplay and fusion of the two in *On Wenxiao Temple*:

> Initially, the people addressed the deity as Jiulang and his temple as Xi Miao 西庙 (West Temple). They held ceremonies of worship at the same time every year. Local officials answered people's wishes and requested the central government to confer the title of Jiulang upon the deity. The central government bestowed the name of Wenxiao Temple upon the temple and the title of *Ying Ji Wang* 英济王 (Wise, Charitable King) upon Jiulang. It was stated in the imperial

decree that the names Jiulang and Xi Miao were not refined and failed to reflect the deity's authority, which in turn would fail to grant the people peace of mind . . . The people had no records for examination and claimed no rules governed their practices. Yet it is undeniable that an emperor by the title of Zhaoming was enshrined there and that the title *Ying Ji* 英济 (Wise, Charitable) was inscribed in the temple. (Yanping Huang 1986, 4.781)

The names "Jiulang" and "Xi Miao" that had been long used by the local followers were deemed "unrefined" by the authorities. In response to the sentiment of the people, the central government conferred titles upon the deity under the national system of rites and bestowed him the titles of "Ying Ji Wang 英济王 (Wise, Charitable King)" and "Wen Xiao 文孝 (Learned, Filial)". However, when it came to the practice of the folk religion, "no rules governed their practices". On the one hand, there was a proactive diffusion of national ideology into the realm of folk religion through the policy of conferring titles upon deities; on the other hand, the implicit acceptance of no records for examination and no rules governing the practice of folk religion carved out a space that allowed the continual folk practice of worshiping Jiulang.

The religious dichotomy between Zhaoming and Jiulang has existed for a long time since the Song dynasty. The identity of "Zhaoming" was accentuated in official rites and academic discourse, documented in literature such as regional texts, while the image of "Jiulang" was presented indirectly at the grassroots. *Shidaixian Zhi* 石埭县志 (*Shidai County Annals*) compiled during the Jiajing reign (1522–1566) of the Ming dynasty, documented the religious dichotomy of "rites vs. customs" and "official authorities vs. the people" clearly: Crown Prince Zhaoming is worshiped at Wenxiao Temple. The day of the sacrificial ceremony is officially documented as the fifteenth day of the eighth month. The people of the county perform their own ceremonies. Due to the great service that the deity has given to Chizhou, its temples are found in every settlement (Yingxiu Huang 1556, vol. 3).

The officials viewed the deity as an ancient sage and an ancestral spirit, thus including the worship of the deity in official ceremonial rites. At the same time, the "great service that the deity has given to Chizhou" and his miraculous feats permeated popular discourse and led to the erection of temples across the settlements and private sacrificial rituals performed by the people. An entry in "Xixi Zhaoming Temple" 溪西昭明庙 in She County 歙县, Huizhou documented: He is also known as Gaozong of Liang. The people refer to him as Guoxi Jiulang. A temple for Jiulang has been erected in Guoxi, Chizhou. It is also a provisional ancestral temple (S. Wang 1982, vol. 5). Similar records exist in abundance. They are birthed in academic narrative and records that documented temples such as Wenxiao Temple or Zhaoming Temple and the deity as Crown Prince Zhaoming or Emperor Zhaoming. In the meantime, the common believer continued to call them Guoxi Jiulang and Jiulang Temple.

The differing representations of Jiulang and Crown Prince Zhaoming are aligned with different religious beliefs and forms of sacrificial ceremonies. Crown Prince Zhaoming's "learnedness and filial piety" as well as his "compassion and virtue" were often emphasized during conferments of titles during the Song dynasty as well as numerous records, prayers for good weather and rain, and poetry written during subsequent eras. Since the Song dynasty, the temples found across the region have been referred to as "Zhaoming Temple", "Wenxiao Temple", "Wenxiao Shrine" and the like. Local officials also utilized diverse means to strengthen the deity's depiction as an ancestral spirit. The magistrate Yuan Shuoyou published the *Selections of Refined Literature* and *Zhaoming Wenji* 昭明文集 (*Zhaoming's Selections of Refined Literature*) in Chizhou and asserted that "Deity and Man are interdependent. The official who accords the deity respect will receive the same in kind. Thus, the deity gets his sacrifice and the official gets his stipend. Neither will suffer from guilt (Yuan 1986, vol. 19, p. 386)." During the Ming and Qing eras, there were several sights of worship found clustering around the Guoxi Temple in Chizhou (Zhaoming Temple). The portrait of Emperor Wu of Liang, Noble Consort Lady Ding, Emperor Jianwen, Emperor Yuan, Consort Lady Cai, the kings of Yuzhang and Zhijiang and their

subjects—officials and personalities such as Shi Baozhi and Fudaishi—could be found in separate locations. The people worshiped and made sacrifices to all of these figures (Lang 2002, vol. 3, p. 666). Historical figures who were associated with Crown Prince Zhaoming were enshrined in the temple, and the manner of sacrificial rituals was designed to accentuate the representation of the deity as an ancestral spirit while erasing the image of Jiulang in folk religion.

Records on "Jiulang"—albeit written by scholars—emphasized various supernatural acts that included preventing calamity and misfortune as well as ending floods and droughts. When documenting supernatural acts in *Zhongjian Zhaoming Taizi Dian Ji* 重建昭明太子殿记 (*On Rebuilding Crown Prince Zhaoming Hall*) during the Ming dynasty, Fang Mo 方谟 wrote, "Sometimes, merchants carrying thousands of *hu* 斛 (unit) of rice enter his domain and are stopped by turbulent waters. The deity calls himself Xiao Jiulang from Guoxi, Chizhou." Similar accounts may be found in the text (C. Wang 1982, vol. 9). Fang's term of address for the deity—"Xiao Jiulang"—reveals that, at that juncture, the deity shared Crown Prince Zhaoming's family name while preserving his name and identity of "Jiulang". During the Ming and Qing eras, the deity's title as "Crown Prince Zhaoming" had gained prevalence while the use of the deity's original title "Jiulang" had declined. Nevertheless, representations of the deity as "Jiulang" could still be seen in welcome ceremonies and in rural villages. Sacrificial ceremonies continued to be suffused with strong Wuism elements. It was written in *Xinghuacun Zhi* 杏花村志 (*Xinghua Village Annals*): Every eighth month of the year, relevant prefectural and county officials would pay their respect and perform ceremonial rites. On the twelfth day of the eighth month, many people would don masks, give performances of "stabbing their arms with weapons" and "piercing their necks with blades", and parade with ceremonial flags and weapons in an elaborate procession, as they were the Seven Sages and Erlang. In the rest of the city and rural villages one to two hundred miles away, clay and wooden dolls for auxiliary deities are placed. Every one of their heads is bowed in the direction of Zhaoming Temple to receive the deity. On that day, officials would offer their respect, then escort Zhaoming to Jingde Temple in a resplendent sedan that resembles an imperial palanquin. The deity's birthday falls on the fifteenth of the month. Jingde Temple is the deity's second home. Up to the eighteenth day of the month, some officials would continue to escort the deity in a similar procession (Lang 2002, vol. 3, p. 666).

Others would offer their respects and perform ceremonies during mid-autumn. This is in accordance with official ceremonial rites for worshiping Crown Prince Zhaoming—an ancient sage and ancestral spirit. The welcome ceremony conducted by the common folk, on the other hand—where followers "don masks, stab their arms with weapons, pierce their necks with blades, and parade with ceremonial flags and weapons in an elaborate procession, as they were the Seven Sages and Erlang"—exhibits strong Wuism characteristics. The mention that "Jingde Temple is the deity's second home" in records suggests a relation between Jiulang and the Buddhist temple, Jingde Temple. Cloaked in the image of Crown Prince Zhaoming, Jiulang—a Wuism deity with elements of the absurd and grotesque—avoided accusations of "heterodoxy" or practicing of "heterodoxy", ensuring its continuing existence and prosperity.

Since the Song dynasty, officials and local scholars reshaped "Jiulang" into the image of "Crown Prince Zhaoming" through academic discourse. By capitalizing on the Song-era ritual policy of conferring titles and inscriptions upon deities, they transformed the deity's representation from that of a Wuism deity "Jiulang" into that of an ancient sage and ancestral spirit "Crown Prince Zhaoming", integrating folk religion under the national system of rites and a shared national identity while advancing regional indoctrination. During the same period, the Wuism characteristics of Jiulang continued to be preserved in rituals such as sacrificial and welcome ceremonies that were conducted by the common people at the grassroots. This implies that in spite of the impact that Song-era national rites continued to have on popular folk religion, the people remained free to continue practicing folk religion. The tension between "rites" and "customs" as well as between the state-recognized deity

and the folk deity is a key reason behind the continuing, long-time coexistence between the historical imperial figure and ancestral spirit "Zhaoming" and the Wuism deity "Jiulang". It is also an intrinsic element within the framework of harmonious coexistence in traditional society.

In reality, the situation of a rites–customs dichotomy is prevalent in the worship of deities in traditional society. *Song Huiyao Jigao: Cimiaomen* 宋会要辑稿·祠庙门 (*Edited Compilation of Song Compendium: Temples*) is the most extensive historical anthology on deities across the country. The compilation covered over 1,200 temples for deities across the country and cited deities—ranging from those that governed the mountains, seas, springs, lakes, reservoirs, ponds, ancient sages and martyrs—who were conferred imperial titles and inscriptions (Xu 2014, 禮 20–21, pp. 987–1108). In particular, conferments of titles and inscriptions by the state upon heterodox deities with fantastical, absurd features and rich Wuism characteristics, threw a cloak of legitimization over them. Similar to the religion of Crown Prince Zhaoming, while such deities' original titles and ceremonies of worship were not recognized by the state, they continue to be used by the people in their everyday practice of religions.

## 5. The Geographical Expansion of the Religion of Crown Prince Zhaoming and the Reasons for Its Continuing Prevalence

Over thousands of years, the religion of Crown Prince Zhaoming existed—from the deity's earliest depiction as the Wuism deity "Jiulang" to his extended dual identity of "Jiulang" and "Zhaoming" since the Song era—and left a significant impact on the common populace across a vast region. Two of the deity's most famous temples—Guoxi Temple and Xiushan Ancestral Temple—were located in Guichi, the region where the religion witnessed its rise. During the Ming and Qing eras, the two temples underwent renovations and repairs, which reflected the continual impact that the religion had on the people in the region. In addition, the religion of Crown Prince Zhaoming spread through all counties in Chizhou. Temples of varying sizes could be found throughout rural areas during the Ming and Qing dynasties. In Qingyang County 青阳县, "a temple could be found in every settlement, and the townspeople of Lingyang were especially devout believers," wrote Xiao Wenbo 萧文伯 in *Chongxiu Wenxiaomiao Ji* 重修文孝庙记 (*On Rebuilding Wenxiao Temple*) during the Ming era. Xiao claimed that at the Jiulang Mound 九郎墩 east of Qingyang is "a temple for Crown Prince Zhaoming of Liang that has stood there since the Five Dynasties period." However, he also stated that Zhaoming was the ninth son of Emperor Wu of Liang and hence named Jiulang 九郎 (Ninth Son) (Cai 1594, vol. 6). "At the foot of Mount Sanfeng 三峰山, in the village of Tang, Guichi, are temples for Yi Lang 一郎 (First Son) and Er Lang 二郎 (Second Son). They are temples of Crown Prince Zhaoming and auxiliaries to the primary temple at Xiushan (Gui 1998, vol. 3, p. 52)." These demonstrate that the people at the grassroots continued to refer to the deity as "Jiulang".

The religion of Crown Prince Zhaoming (Jiulang) originated in Chizhou and spread throughout the coastal region of Wannan 皖南 early on. In the early Song dynasty, the prevalence of the religion of Jiulang in Jianghuai was documented in *New Southern Anecdotes*. *Song Compendium* gave an explicit description: "Crown Prince Zhaoming" was received from Chizhou and in Xuancheng during the Zhiping reign (1064−1067) (Xu 2014, p. 1087). Numerous temples for Crown Prince Zhaoming (Jiulang) were found throughout the counties in Huizhou during the Qing and Ming eras. In Qimen County 祁门县, "there are four Wenxiao Temples built for worshiping Gaozong of Liang. One is in the village of Cao, Wudu; one in the village of Zhou, Wudu; one in Shanhe, Liudu; one in Baixi, Badu" (S. Wang 1982, vol. 5). This shows that the worship of Crown Prince Zhaoming was prevalent in many towns and villages. In Yi County 黟县, "Wenxiao Temple is found five miles southwest of the county. The temple was built for worshiping Crown Prince Zhaoming, Xiao Tong." "The people today call it Gaozong Temple (Cheng and Yu 1998, vol. 11, p. 363)." "There is a Xixi Zhaoming Temple in She County 歙县 for Gaozong of Liang. The people refer to him as Guoxi Jiulang. A temple for Jiulang has been erected in

Guoxi, Chizhou. It is also a provisional ancestral temple (S. Wang 1982, vol. 5)." A conclusion can be thus drawn from this that the religion of Crown Prince Zhaoming (Jiulang) has penetrated the grassroots of society in Wannan, playing quite a significant role in religion and tradition as well as having a strong influence on the people.

During the Song dynasty, Raozhou 饶州, Xinzhou 信州 and Nankang Military Prefecture 南康军—among other areas such as Chizhou—were under the administrative jurisdiction of Jiangnan East Circuit and shared a strong relationship with Hongzhou 洪州, a key settlement under the administrative jurisdiction of Jiangnan West Circuit. The religion of Crown Prince Zhaoming (Jiulang) entered these regions early on. Many temples for the religion could be found across the regions up until the Ming and Qing eras. In Poyang County 鄱阳县, Raozhou, "Guoxi Temple stands in front of the old prison at Yongping Gates, and it is a temple for Crown Prince Zhaoming of Liang. During the Song dynasty, there is a temple for a guardian against fires. The common folk would travel to Chizhou to receive and pay their respects to the deity." Records clearly showed that the deity enshrined in the temple was received from Chizhou and that many repairs and renovations to the temple were carried out event in the late Qing (Y. Wang 1682, vol. 4). A temple was built in Leping County 乐平县 as early as the Song dynasty. "During the Jiaxi reign (1237–1240), the magistrate Luo Lu 罗侣 renovated the temple. A fire broke out at the West Gates in the third year of the Chunyou reign (1241–1252). The deity granted aid by weakening winds and putting out the fire." During the Qing era, the temple was destroyed and rebuilt repeatedly (M. Chen 1870, vol. 2). In Qianshan County 铅山县, Xinzhou, "Guoxi Temple is located two hundred and fifty steps west of the county. The temple was constructed to worship Emperor Wenxiao from Xiushan, Chizhou, also known as Crown Prince Zhaoming of Liang". The temple was originally built during the Xuanhe reign of the Northern Song dynasty, then renovated during the Shaoding reign of the Southern Song dynasty. The religion continued to flourish until as late as the Qing era (Hua 1873, vol. 6). The Wenxiao Temple in Xingzi County 星子县, Nankang "is located at the region's western bay. The temple is built to worship Crown Prince Zhaoming of Liang, Xiao Tong (L. Chen 1982, vol. 7)." The Wenxiao Temple in Xinjian County 新建县, Hongzhou was built along the river outside Zhangjiang Gates (Huang Zhang 1985, vol. 10). Among local temples in Linjiang Prefecture 临江府, one was called Emperor Wenxiao Temple. "Local people made ritual sacrifices in local temples, in the event of droughts and illness." The local populace worshiped this deity (S. Liu 1982, vol. 8).

From the Song dynasty to the Ming and Qing dynasties, the religion of Crown Prince Zhaoming (Jiulang) was also practiced in many areas in Zhejiang. A temple was built in Nanjing as early as the Song dynasty. "Wenxiao Temple is built to worship Crown Prince Zhaoming of Liang. It is situated in the southwestern part of the city, west of Xinqiao, and faces Huaishui River. The building was destroyed and rebuilt in the fifth year of the Shaoxing reign (Y. Zhou 1990, vol. 44, p. 2057)." The Wenxiao Temple outside Wuhu's West Gates "is built for worshiping Crown Prince Zhaoming of Liang, also known as Emperor Wenxiao (Yue Huang 1999, vol. 35, p. 663)." The Wenxiao Temple in Taizhou 泰州 also finds its origins in Chizhou. *Taizhou Zhi* 泰州志 (*Taizhou Annals*) documented that "Wenxiao Temple is located west of the prefecture and built in the thirteenth year of the Shaoxing reign of the Song dynasty. The people referred to the deity enshrined in the temple as Guoxi Jiulang and worshiped him as a deity of fire (S. Chen 1998, vol. 12, p. 89)." The difference is that the deity was referred to as a deity of fire in Taizhou. A Wenxiao Temple was also built at Mount Gui 龟山, Chongfeng Village 崇奉乡, Ruian County 瑞安县, Zhejiang (Wang and Cai 1990, vol. 16, p. 740).

The people relied on the deified Crown Prince Zhaoming (Jiulang) for survival and pinned their psychological hopes on the deity. That is the key reason behind the religion's widespread proliferation and presence. The fear and powerlessness that the people experienced in the face of diseases, plagues, disasters and misfortunes drove them to prayer and to deities, from whom they seek faith and hope. These emotions fueled the never-ending emergence and widespread existence of folk religions in traditional society (Y. Li

2004, p. 21). "As the people worship the deity, so does the deity bless Chizhou. They make offerings to the deity when they eat and drink; they pray to the deity when there are floods, droughts, or diseases. Their prayers are always answered (C. Wang 1982, vol. 9)." The attribution of miraculous feats to Crown Prince Zhaoming (Jiulang) since the Song dynasty, reflects his believers' physical and psychological reliance on the deity. Tales of such feats also sustained the religion and enabled its continuing prevalence. Crown Prince Zhaoming (Jiulang) was primarily viewed as a deity of water that brings rain or good weather and a guardian of ships and boats. "The four areas of Guichi have thirty-six *bao* (保, an administrative unit, composed of ten households). When some *baos* were hit by a drought, local people came to the temple in the West to collect water from its well. Each carried a willow tree branch to prevent others from robbing them of their water before they could ascend the altar and pray. Rain fell as soon as their prayers were uttered. It was Zhaoming who blessed them with rain (Lang 2002, vol. 12, p. 755)." To answer the myriad needs of his followers, Crown Prince Zhaoming (Jiulang) gradually became an all-encompassing divine guardian who prevented calamity and misfortune and provided aid in times of trouble. Neo-Confucianist Wei Liaoweng 魏了翁 once offered sacrifices and prayed for protection and aid to end the local rebellion, praying: "To the deity who prevents calamity and misfortune in exchange for sacrifices. Let the foolish and mad destroy themselves. Take our rebels and consolidate our lands. His Majesty (the emperor) is enraged and has given me charge over the country to station troops in Chiyang 池阳 and win respect and subservience through military means. Please watch over my expedition, let me exterminate the malicious rebels, and bring peace to the country. Please manifest your divine spirit (Wei 1986, vol. 98, p. 755)." In conclusion, the fear and powerlessness that the people experienced in the face of diseases, injuries, plagues, disasters and misfortunes as well as their desire for peace, for mild winds and sufficient rainfall, and for the absence of diseases and disasters, have led to their praying to and placing their faith in Crown Prince Zhaoming (Jiulang). This is the key reason why the religion continued to flourish.

Merchants played a crucial role in the spread of the religion to many areas in Jiangxi. The most prominent divine duty of Crown Prince Zhaoming (Jiulang) was the protection of ships and boats. The presence of temples along the waterways between the Yangtze River and Poyang Lake had a close connection with merchants. To ensure that their prayers were answered, merchants in Jiangxi, Raozhou resorted to stealing statues from the Guoxi Temple in Chizhou. "The statue's head was stolen by a prominent merchant in Jiangxi. According to Raozhou's official records, a merchant from Chizhou requested the return of the statue's head during the Jiajing reign. A temple was thus built outside the East Gates and named Guoxi so that the origins of the deity would never be forgotten (Lang 2002, vol. 12, p. 755)." The street outside the Zhangjiang Gates of Nanchang, Jiangxi was named Wenxiao Temple Street because of the Wenxiao Temple located there. West of the temple stood the storehouse for grains transported via the waterways. "Grains imported every year are delivered here (Q. Chen 1870, vol. 3)." The location of the temple suggested that the merchants might have constructed the temple to seek protection and divine blessing for their deliveries. The proliferation of the religion of Crown Prince Zhaoming to Ruian, Wenzhou, Zhejiang might have something to do with the deity's divine duty as a guardian of merchants and maritime voyages. In the Wenzhoufu Zhi 温州府志 (*Wenzhou Prefecture Annals*) complied during the Hongzhi reign of the Ming dynasty, it was written, "During the Song era, fishermen at sea spotted incense smoke amidst the waves, and they believed that to be the manifestation of Crown Prince Zhaoming of Liang. Thus, the deity was enshrined at Mount Gui (Wang and Cai 1990, vol. 16, p. 740)."

An effective organizational structure for sacrificial rituals is a key reason behind the continuing prevalence of the religion of Crown Prince Zhaoming aka Jiulang. The religion had a quite stable organizational mechanism where one clan was responsible for performing sacrificial rituals across many generations over thousands of years. This is a significantly rare phenomenon. Song-era records stated that "since the Kaicheng reign of the Tang dynasty, the charge over sacrificial rituals" at Guoxi Temple in Chizhou had belonged

to the Zhou clan. During the Southern Song dynasty, "the descendants of the Zhou Clan split into eight factions, for the best." The Zhou clan remained in charge of sacrificial rituals during the Ming and Qing eras. Zhou Liangcheng 周良诚 was the temple attendant during the Hongwu reign of the Ming dynasty. The imperial decrees conferring titles and inscriptions during the Song dynasty were "destroyed by the Zhou clan on the seventh year of the Tianqi reign of the Ming dynasty". Up until the Qing era, the Zhou clan still resided close to the temple and took charge of the sacrificial rituals. However, the clan appeared to have suffered a decline. "The Zhou clan occupies the temple's farmland but does not perform the sacrificial rituals in spring and autumn. They are in decline, their initial huge numbers dwindling to but a few (Lang 2002, vol. 3, p. 667)." The *Chizhoufu Zhi* 池州府志 (*Chizhou Annals*) of the Qianlong reign of the Qing dynasty still documented "the Zhou clan has been in charge of the sacrificial rituals since the Kaicheng reign of the Tang dynasty and owns four *qing* 顷 (unit) of temple's farmland (Shifan Zhang 1998, vol. 18, p. 297)." The reasons behind how the Zhou clan could retain its power across generations and over a millennium as well as the reasons behind its decline are worth examining in further detail. Besides intergenerational control over sacrificial rituals, the comprehensive nature of the religion's organizational mechanism is revealed in its sacrificial ceremonies such as the welcome ceremony. Take the ceremony performed in Shidai, Chizhou, where "the top twenty families receive Zhaoming at the altar and perform sacrifices every fifteenth day of the eighth month (T. Zhou 1675, vol. 3)." Every temple had a leader who would get the people together to perform the sacrificial rituals. The organizational mechanism penetrated the deepest level of society, becoming a part of society at its grassroots and affecting how the grassroots operated. Folk religion also became a part of people's everyday life.

Economic factors motivated people to organize religious ceremonies such as Wuism and other sacrificial rituals. They were also one of the reasons behind officials' involvement in folk religion. The organizers of sacrificial rituals took advantage of such opportunities for personal gains. "At small events, they take the sacrifices of chickens and pigs back home; at big gatherings including dancing, they take away the remaining sacrificial offerings (Xu 2014, p. 992)." The expenses for sacrificial rituals were enormous. "One person's costume may cost up to one to two hundred. The cheapest is over twenty. This has always been the case (Lang 2002, vol. 9, p. 734)." Guichi was the traffic hub for transport on the Yangtze River while Guoxi Temple was where all vessels passed. Vast crowds made offerings at the temple and drew the covetous eyes of local officials. In *Mizhai Biji* 密斋笔记 (*Mizhai's Journal*), it was written: "The temple for Zhaoming of Liang in Chizhou has accumulated money. The officials deceived the people and lied that they wished to collect money and use it to request the government to confer a title upon the deity. The officials of Tongling took such money of nearly fifty thousand. The magistrate suddenly puked blood and died. Both officials were impeached (Xie 2015, vol. 5, p. 164)." The account sheds light on why local Song-era officials were so eager to advocate for conferments of titles for deities. While part of the reason might be attributed to their responding to public sentiment and their duty toward local indoctrination, another part was inevitably motivated by financial gains.

The dichotomous coexistence of a state-recognized deity and a folk deity played a part in the lasting prevalence of the religion of Crown Prince Zhaoming. In discussing "rites", special attention should be given to the role that the literati played in promoting rites during the Ming and Qing eras. The influence that the historical figure Xiao Tong and his *Wenxuan* 文选 (*Selections of Refined Literature*) resulted in a vast library of poems and essays by local officials and literati as well as their financial support for the renovation of temples during the Ming and Qing dynasties. This played an important part in the spread of the religion of Crown Prince Zhaoming. Take the Xiushan Temple, Chizhou, for example, which rose to prominence again during the Ming and Qing eras. Related literary works and texts about the temple abound. These works were compiled in *Xiushan Zhi* 秀山志 (*Xiushan Annals*) during the Qing dynasty. In *Xiushan Annals*, it was written that Xiushan Temple "was built in the third year of the Datong reign of the Liang dynasty. Sacrificial rituals have been held at the temple since the Chen and Sui dynasties." "In the third year

of the Datong reign of the Liang dynasty, the tomb of crown prince was built at Shicheng 石城 at the request of local people"; etc. (H. Chen 1994, vol. 5, p. 333). Such records are absent from Song-era texts such as *Facts About Crown Prince Zhaoming*, which demonstrates that they were interpretations by local Qing and Ming-era scholars. Many reasons prompted the common populace to pray, sacrifice to and house the deified Crown Prince Zhaoming in their hometown. The obvious one is that the deity answered their prayers. In the eyes of local officials and scholars, the deity that the common people housed in their hometown and offered sacrifices to was Crown Prince Zhaoming, the figure renowned for being learned, virtuous, compassionate and filial. They had no reason to ban the practice of the religion. In fact, they seized the opportunity, responded to public sentiment and fulfilled their task of indoctrinating the public. Temples were referred to as Wenxiao Temples, Zhaoming Temples, and so on throughout Wannan, Jiangxi and Zhejiang. Narratives that seem to lack historical truth on the one hand—"Zhaoming was the ninth song of Emperor of Liang", for example—are an accurate reflection of the connection between the folk deity and the state-recognized deity. The rites–customs dichotomy has given the religion of Crown Prince Zhaoming greater room for existence and is one of the key reasons behind the religion's widespread proliferation and continuing prevalence.

## 6. Conclusions

After Kristofer Schipper highlighted the differences between Taoist priests in Taoism and folk religions, Edward L. Davis further differentiated the differences between Taoist priests, Buddhist tantric priests, and Wuist shamanistic priests in his work *Society and the Supernatural in Song China*. The subjects of "great tradition vs. little tradition" and "nation vs. region" became a focal point of many previous studies on folk religions. However, these studies did not delve into how great tradition or national identity entered the domain of little tradition or folk religions. By focusing on the interplay between "rites" and "customs", this study reveals how "rites", which represent the national identity, through the discourses of government officials and scholars, and the national system of conferment of titles and inscriptions, permeated and transformed folk religions while allowing space for the original "customs" in folk religions to survive. Based on the study, we know that such a coexistent relationship is prevalent in Chinese folk religions.

In this paper, the author endeavored to study how the folk or heterodox deity Jiulang was transformed into the state-recognized or orthodox deity Crown Prince Zhaoming. The objective of this study is to explore the interactive relationship between folk deities and state-recognized deities as well as how such a relationship influenced the propagation of folk religion. In the case study of the religion of Crown Prince Zhaoming, both the image of the Crown Prince deity legitimized by the state coexisted with the image of Jiulang recognized by the masses in a prolonged rites–customs dichotomy, without coming under attack from the authorities for heterodoxy and Wuism. In addition, the religion of Crown Prince Zhaoming retained the beliefs that the common populace held Jiulang as a deity that answered all prayers. These factors contributed to the widespread proliferation and lasting prevalence of the religion.

First of all, the religion of Crown Prince Zhaoming has its roots in the heterodox religion of Jiulang. During the Tang era, the religion of Jiulang was already enjoying some level of popularity in Chizhou and had a relatively established structure of organized sacrificial rituals. It was relatively popular in the Jianghuai region during the Five Dynasties and early Song periods. Alongside the religion of General Mao, the religion of Jiulang was recognized as heterodoxy that proliferated among the general populace. At that point, Jiulang was still not associated with Crown Prince Zhaoming.

Secondly, the transformation of folk Jiulang into the state-recognized Crown Prince Zhaoming in the Song dynasty took place against the backdrop of the Song authorities' reformation of local folk religions. Local officials with a duty to educate the masses and local scholars who wished to build a local culture and traditions, through their writings on temples (庙记), associated the historical figure Crown Prince Zhaoming with the folk deity

Jiulang, thereby creating the identity of Jiulang as a state-recognized deity called Crown Prince Zhaoming. The Song dynasty conferred titles and inscriptions upon folk deities under the national system of rites. Jiulang received seven titles and inscriptions between the third year of the Yuanyou reign (1088) and the first year of the Jiatai reign (1201), and thus it was legitimized by the stated deity Crown Prince Zhaoming. It was through the reshaping of religious discourse by national ideology as well as the infiltration of national rites into the realm of folk religion, that the folk deity Jiulang was transformed into the state-recognized deity Crown Prince Zhaoming.

Thirdly, the state-recognized deity Crown Prince Zhaoming and folk deity Jiulang co-existed in a prolonged rites–customs dichotomy. Through national discourses espoused under official rites, Jiulang was turned into Crown Prince Zhaoming, a deity of imperial lineage. Meanwhile, in the actual practice of folk religion, the deity Crown Prince Zhaoming continued to persistently retain its image as the folk deity Jiulang while sacrificial rites beyond official rituals continued to be held in his name. This coexistent dichotomy between "rites" and "customs" as well as the state-recognized deity and the folk deity represents the normative state of folk religions in traditional society in China.

Fourthly, many factors contributed to the widespread proliferation and popularity of the religion of Crown Prince Zhaoming in Jianghuai and Jiangnan. The people's fear and powerlessness against diseases, plagues, disasters and misfortunes led to their praying and hoping for peace and safety. This is the key reason why the religion of Crown Prince Zhaoming continued to thrive. Merchants played a key role in the spread of the religion across Jiangxi. The Zhou clan was in charge of temple rites over many generations, facilitating the establishment of an organized religious structure. The state-recognized deity and the folk deity coexisted in a dichotomy relationship, and both the authorities and the common populace contributed to the enduring popularity of the religion of Crown Prince Zhaoming.

From the Song dynasty, the national system of rites permeated and impacted upon folk religions through official and academic discourses, propelling the latter's continuing transformation into an orthodox form. Nevertheless, there remained spaces between the national system of rites and folk religions and traditions where "rites" and "customs" interacted, integrated and coexisted well. The interplay, fusion and peaceful coexistence between "rites" and "customs" is the normative state of folk religions in traditional society in China. Furthermore, this paper's case study on the religion of Crown Prince Zhaoming noted the Zhou clan's charge over the worship of the deity from the Tang to the Ming and Qing eras, demonstrating that such intergenerational succession is rarely documented in the practice of a folk religion and is thus worthy of note.

**Funding:** This research was funded by [Study on Chinese Ritual Culture and National Governance] grant number [22VLS004].

**Institutional Review Board Statement:** Not applicable.

**Informed Consent Statement:** Not applicable.

**Data Availability Statement:** Not applicable.

**Conflicts of Interest:** The author declares no conflict of interest.

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
