# Peer review of "The Interplay of Rites and Customs: The Evolution and Regional Propagation of the Religion of Crown Prince Zhaoming"

_religions, doi:10.3390/rel14050678_

Round 1
Reviewer 1 Report
This is a very interesting article with a rich discussion about how the state-recognized deity and religion of the Crown Prince Zhaoming emerged from folk traditions surrounding Jiulang, a Wuism deity. In addition to its many textual references, the research incorporates fascinating examples of material culture to support the thesis. Overall, the writing is very good.
In lines 95-97, the author states: “The objective of this paper is to straighten out the origins of the religion of Crown Prince Zhaoming.” A similar statement should be made right in the paper’s thesis paragraph at the beginning of the Introduction. The first paragraph of the Intro includes important supportive material. However, the Crown Prince Zhaoming should be mentioned very near to the top of the paragraph and Introduction.
Some important works that the author might want to reference include Vincent Goossaert’s article, “The Concept of Religion in China” and Edward Davis’ book, “Society and the Supernatural in Song China.” Both of these works discuss how folk religions in China have been studied or not studied, including as pertains to Song China.
Some cases of passive voice should be removed. For example, in line 12, the author states, “the central place for the worship of Jiulang was shifted from the ancestral temple…” when the sentence would be better with the passive voice removed and changed to: “the central place for the worship of Jiulang shifted from the ancestral temple…” A similar case appears in lines 249-250. Please check the document for instances of passive voice.
For the author or copyeditor: definite articles should be added in front of some of the names of literature (e.g. “the” Geographical Anthology [line 298] and “the” Universal Anthology in the Taiping Reign [line 300]). Please double check the text for these. Line 398 should state “the state-recognized deity” (currently missing the article “the”). Also, “As early as the early Song dynasty” is repetitive in line 615 and should be changed.
Author Response
Dear reviewer:
I thank you for the critical comments and helpful suggestions. I have taken all these comments and suggestions into account, and have major correction in this revised manuscript. The details are as follows:
This is a very interesting article with a rich discussion about how the state-recognized deity and religion of the Crown Prince Zhaoming emerged from folk traditions surrounding Jiulang, a Wuism deity. In addition to its many textual references, the research incorporates fascinating examples of material culture to support the thesis. Overall, the writing is very good.
- In lines 95-97, the author states: “The objective of this paper is to straighten out the origins of the religion of Crown Prince Zhaoming.” A similar statement should be made right in the paper’s thesis paragraph at the beginning of the Introduction. The first paragraph of the Intro includes important supportive material. However, the Crown Prince Zhaoming should be mentioned very near to the top of the paragraph and Introduction.
Response 1: Thank you for your suggestions! Yes, my objective was to reveal the interplay between “rites” and “customs” in folk religion by examining the evolution of the religion of Crown Prince Zhaoming. I agree this should be made clear at the beginning of the article. I have amended the first paragraph of the Introduction accordingly. Similarly, I have also made some revisions and additions to the literature review in the Intro. Please refer to pages 2 and 3 for details.
- Some important works that the author might want to reference include Vincent Goossaert’s article, “The Concept of Religion in China” and Edward Davis’ book, “Society and the Supernatural in Song China.” Both of these works discuss how folk religions in China have been studied or not studied, including as pertains to Song China.
Response 2: Thank you for your suggestions! They are important works pertaining to this field. Edward Davis’ Society and the Supernatural in Song China, especially, demonstrates the renewed vitality of the Song dynasty’s Taoist, Buddhist, and local religious traditions, providing great insights for the study in this paper. I have included both works in the literature review. Please refer to page 2 for the revisions.
- Some cases of passive voice should be removed. For example, in line 12, the author states, “the central place for the worship of Jiulang was shifted from the ancestral temple…” when the sentence would be better with the passive voice removed and changed to: “the central place for the worship of Jiulang shifted from the ancestral temple…” A similar case appears in lines 249-250. Please check the document for instances of passive voice.
Response 3: Thank you for your comments on the use of passive voice in the article. They are very helpful in raising the overall quality of the writing. I have edited the parts that you have highlighted above as well as checked the entire article for instances of passive voice. Please refer to pages 1 and 6 for the revisions.
- For the author or copyeditor: definite articles should be added in front of some of the names of literature (e.g. “the” Geographical Anthology [line 298] and “the” Universal Anthology in the Taiping Reign [line 300]). Please double check the text for these. Line 398 should state “the state-recognized deity” (currently missing the article “the”). Also, “As early as the early Song dynasty” is repetitive in line 615 and should be changed.
Response 4: Thank you for your comments. They are very helpful in raising the overall quality of the writing. I have edited the parts that you have highlighted above as well as checked the entire article for instances of passive voice. Please refer to pages 5, 8, 13, 16, and so on for the revisions.
Thank you again for your criticism and suggestions

Reviewer 2 Report
The research topic is good. This article tries to fill the research gap on the religion of crown prince zhaoming.
Few comments to the author:
1. The literature review in the introduction section is good. However, most of the references are in Chinese work. I suggest the author include Western references in the related Chinese folk beliefs/religions.
2. The abstract is too long. The author should simplicate it:
point out a clear research problem on this topic
summarize his/her final research finding/significance
3. The main text's elaboration on Crown Prince Zhaoming's revolution is quite details. But I think the main sections in the article are rich in content but lack self-opinion and analytics. I hardly find the core thesis in the article. How important the case study could reflect the ancient Chinese folk religions. What are the contributions of this work to the recent scholarships? Does it engage with recent research methodology in this research field? What is so unique about this case study?
the language is fine
Author Response
Dear reviewer:
I thank you for the critical comments and helpful suggestions. I have taken all these comments and suggestions into account, and have major correction in this revised manuscript. The details are as follows:
The research topic is good. This article tries to fill the research gap on the religion of crown prince zhaoming. Few comments to the author:
- The literature review in the introduction section is good. However, most of the references are in Chinese work. I suggest the author include Western references in the related Chinese folk beliefs/religions.
Response 1: Thank you for your suggestions! A comprehensive understanding of previous studies conducted and their findings is pertinent to carrying out further studies. Many relevant research achievements by Kristofer Schipper, Edward L. Davisand other scholars are instructive. I have included relevant Western references that would be helpful in the article. Please refer to pages 1 and 2 for details.
- The abstract is too long. The author should simplicate it: point out a clear research problem on this topic summarize his/her final research finding/significance.
Response 2: Thank you for highlighting the deficiency and providing a helpful solution! I agree that the abstract should be refined so that the article’s theme is clear. I have also emphasized and expanded upon the findings and significance of this study. Please refer to page 1 for details.
- The main text's elaboration on Crown Prince Zhaoming's revolution is quite details. (1)But I think the main sections in the article are rich in content but lack self-opinion and analytics. I hardly find the core thesis in the article.
Response 3: Thank you for highlighting the deficiency! I believe that it is necessary to reconstruct a full picture of the origin, evolution, and spread of the religion of Crown Prince Zhaoming with a comprehensive and detailed exploration of existing historical records and literature. I have consistently highlighted the key theme of the thesis – the interplay of “rites” (national system of rites) and “customs” (the practice of folk religions) – throughout the paper, ended each section with a discussion and critical analysis relating to the said interplay, and summarized my thoughts on the matter in the conclusion. I have revised the article to further highlight the article’s key points and emphasize the key points of my argument in the conclusion. Please refer to page 16 and 17 for details.
(2)How important the case study could reflect the ancient Chinese folk religions. What is so unique about this case study?
Response 4: Thank you for your comments on the significance of the case study! In southern China, all kinds of sorcerer rituals with exorcism as the core and a whole set of related religious culture are closely related to the folk witchcraft culture tradition. In the course of its development, they are also influenced by the so-called "orthodox" religions, "Confucianism, Buddhism and Taoism" and constantly transformed.
Crown Prince Zhaoming is a prominent historical figure, well known for his literary achievements and filial piety. The Crown Prince Zhaoming depicted in historical records is the near-perfect embodiment of “rites”. The practice of the folk religion of Crown Prince Zhaoming, on the other hand, often goes against and is inconsistent with the tenets of official “rites”. Based on this study, we know that these discrepancies stem from the original source from which the religion of Crown Prince Zhaoming evolved – the Nuo deity Jiulang. Simply put, the evolution of the religion of Crown Prince Zhaoming from Tang and Song dynasties until today is a classic case of the interplay between “rites” and “customs”. The case study has great significance in the study of Chinese folk religions, and the interplay between “rites” and “customs” shown in the religion is prevalent in other Chinese folk religions. This is why this case study is meaningful and significant.
(3)What are the contributions of this work to the recent scholarships? Does it engage with recent research methodology in this research field?
Response 5: Thank you for your comments on the study’s contribution and methodology! After Kristofer Schipper highlighted the differences between Taoist priests in Taoism and folk religions, Edward L. Davis further differentiated the differences between Taoist priests, Buddhist tantric priests, and Wuist shamanistic priests in his work Society and the Supernatural in Song China. The subjects of “great tradition vs little tradition” and “nation vs region” became a focal point of many previous studies on folk religions. However, these studies did not delve into how great tradition or national identity entered the domain of little tradition or folk religions. By focusing on the interplay between “rites” and “customs”, this study reveals how “rites”, which represent the national identity, through the discourses of government officials and scholars, and the national system of conferment of titles and inscriptions, permeated and transformed folk religions while allowing space for the original “customs” in folk religions to survive. Based on the study, we know that such a coexistent relationship is prevalent in Chinese folk religions.
In terms of methodology, a combination of research methods was adopted in this study. Firstly, I conducted an empirical study on the existing rich repository of historical records to reconstruct the origin, evolution, and regional transmission of the religion of Crown Prince Zhaoming. Next, I relied on the research method common to religious studies and historical anthropology, with a focus on interpreting existing literature by scholars and officials, discerning the factors influencing the folk religion that are hidden in such discourses, and incorporating these findings with further studies on the actual practices of folk religions. Finally, I employed the research method common to the fields of religious communication and historical geography, focusing on the study of space and examining the transmission of the religion of Crown Prince Zhaoming through geographical spaces as well as the factors driving said religious propagation.
Thank you again for your criticism and suggestions.

Round 2
Reviewer 2 Report
I am satisfied with the author's amendment. Thank you so much.